# Physiology: An Important Tool to Assess the Welfare of Aquatic Animals

**DOI:** 10.3390/biology10010061

**Published:** 2021-01-15

**Authors:** Ismael Jerez-Cepa, Ignacio Ruiz-Jarabo

**Affiliations:** 1Department of Biology, Faculty of Marine and Environmental Sciences, International Campus of Excellence in Marine Science (CEI·MAR), University of Cádiz, 11510 Puerto Real, Spain; ismael.jerez@uca.es; 2Department of Animal Physiology, Faculty of Biological Sciences, University Complutense, 28040 Madrid, Spain

**Keywords:** cephalopods, crustaceans, dipnoans, elasmobranchs, fish, physiology, stress, welfare

## Abstract

**Simple Summary:**

Ensuring a good quality of life for animals is a matter of concern. Welfare assessment has been quite well developed for many terrestrial species, but it is less well characterized for aquatic animals. Classic methodologies, such as behavioral observation, seem unable to improve the wellbeing of aquatic animals when used alone, mainly due to the large number of species and the difficulty to obtain comparative results among taxa. For this reason, it is necessary to identify more methodologies that may be common to the main aquatic taxa of interest to humans: Fish, cephalopods, and crustaceans. Here we present a physiological framework for these taxa as a proxy to evaluate aquatic animal welfare. Physiology is a useful tool in this regard, since animals maintain their homeostasis in a range of values determined for each parameter. Changes occur depending on the type and degree of stress to which animals are subjected. Therefore, understanding the physiology of stress can offer information that helps improve the welfare of aquatic animals.

**Abstract:**

The assessment of welfare in aquatic animals is currently under debate, especially concerning those kept by humans. The classic concept of animal welfare includes three elements: The emotional state of the organism (including the absence of negative experiences), the possibility of expressing normal behaviors, and the proper functioning of the organism. While methods for evaluating their emotions (such as fear, pain, and anguish) are currently being developed for aquatic species and understanding the natural behavior of all aquatic taxa that interact with humans is a task that requires more time, the evaluation of internal responses in the organisms can be carried out using analytical tools. This review aims to show the potential of the physiology of crustaceans, cephalopods, elasmobranchs, teleosts, and dipnoans to serve as indicators of their wellbeing. Since the classical methods of assessing welfare are laborious and time-consuming by evaluation of fear, pain, and anguish, the assessment may be complemented by physiological approaches. This involves the study of stress responses, including the release of hormones and their effects. Therefore, physiology may be of help in improving animal welfare.

## 1. Introduction

### 1.1. Welfare

Human–animal interactions are defined by complex and refined relationships. A desire to improve these relationships has led us to be concerned about the conditions of the animals and their wellbeing. Animal welfare is a difficult concept to define, mostly due to anthropomorphic perception, but it is usually associated with three different aspects: (i) The physiological functioning of an animal; (ii) its natural living environment; and (iii) its feelings [1,2]. Traditionally, in farmed animals, welfare has been associated with the fulfillment of the “Five principles of freedom”, described by the Farm Animal Welfare Council (FAWC) [3]. These five principles serve to guarantee the vital needs in the absence of negative experiences. Therefore, animals must be free from hunger, thirst, discomfort, pain, disease, fear, anguish, and must be free to express their natural behavior. This definition is widely general and vague. In this sense, other authors prefer the term “quality of life”, where both negative and positive experiences are assumed in the normal life of any animal, all of them supporting the proper development of individuals [2]. Based on these (and other) definitions, a complete set of legislation has been developed in many countries to guarantee and improve the welfare of animals interacting with humans, as discussed below.

Governmental concern about animal welfare, described in relevant policies, is gaining importance, not only for terrestrial ecosystems but also for aquatic ecosystems. In Europe, the initial regulations were directly related to animal sentience [4], something traditionally considered for mammals, birds, and reptiles, but recently extended to fish [5]. However, due to the lack of sufficient scientific evidence, problems have arisen when determining animal sentience for the entire collection of aquatic animals that are subjected to human influence (i.e., aquaculture, fisheries, zoos, etc.). Nevertheless, the current trajectory is to increase the number of taxa protected by these regulations, including invertebrates such as crustaceans and cephalopods [6]. Legislation tends to regulate all taxa that are kept captive, as well as their procedures of humane slaughtering [6]. Outside Europe, most of the current legislation considers farmed fish as the only aquatic animals that deserve a welfare treatment. Therefore, the “OIE Aquatic Animal Health Code” (World Organization for Animal Health) introduces recommendations to apply to farmed fish during stocking, transport, stunning, and killing [7]. Similarly, the FAO (Food and Agriculture Organization of the United Nations) has recently published a report in which the welfare of farmed finfish is addressed, but excludes animals captured in the wild (commercial or recreational fisheries), and issues related to the culture of crustaceans and mollusks are not considered [8]. In the European Union, the regulations related to welfare in aquaculture are the Council Directive 98/58/EC [9], the Regulation (EC) 882/2004 [10], the Regulation (EC) 1/2005 [11], the Council Directive 2006/88/EC [12], and the Regulation (EC) 1099/2009 [13]. Despite all these regulations, no specific requirements for the husbandry, transport, or slaughter of fish are considered [8]. There is only one exception, the Regulation (EC) 710/2009 [14], in which the rules for organic aquaculture for animal and seaweed production are detailed. This regulation establishes the obligation to guarantee the welfare conditions for fish, crustaceans, mollusks, and echinoderms, and defines the conditions of water quality, stocking density, and feeding regime, without further information about their specific necessities, behaviors, and/or physiological conditions. Regarding fisheries, certification organizations, such as the Marine Stewardship Council (MSC), are starting to consider ensuring animal welfare during capture and slaughter of fish, but this is far from being applied due to the complexity of its evaluation [15]. The current regulation for keeping animals in zoos and aquaria in Europe is the Council Directive 1999/22/EC [16], and only establishes the obligation to maintain welfare conditions in a general and vague way. However, the European Commission [17], and the European Association of Zoos and Aquaria (EAZA) [18], have recently published documents of good practices to improve the standards for the accommodation and care of animals in zoos and aquaria. The most advanced regulations about welfare of aquatic animals are those related to animals used in experimentation [6]. For example, the EU Directive 2010/63/EU [19], which includes fish and cephalopods, considers physiological stress responses as part of welfare assessment during the care and accommodation of animals. Similarly, other countries, such as Australia, Canada, New Zealand, Norway, and Switzerland, include not only cephalopods but also decapod crustaceans in their regulations of animal experimentation [6].

Current approaches to determining whether taxa should be included in animal welfare regulations are based on animal sentience, which is a complex term to apply for some aquatic taxa, especially invertebrates [6]. Some researchers have focused on the neuroanatomical search for nociceptors (sensory neurons that respond to potential damaging, also known as “pain receptors”) [2,20,21]. However, the idea of pain, suffering, and, above all, consciousness, remains to be sufficiently determined for humans, let alone for animals. Therefore, this concept must be applied with caution for legislation [6]. Another large area of study in relation to animal welfare has focused on behavior [22,23]. In terrestrial animals, a list of anomalous behaviors related to discomfort has been developed [23]. This has been possible due to the ease with which we can observe these animals, as well as the (relatively) small number of terrestrial species managed in captivity and in natural environments. However, only (relatively) few aquatic species kept in captivity can be observed directly, since there is an immense variety of taxa, with different shapes, sizes, and biological needs, making it very difficult to describe their behavior adequately. Moreover, the variation in individual fish responses to the same situation (which have been studied extensively for aquaculture purposes), make an overall assessment very difficult [6]. The concept of homeostasis is well known in animals as the equilibrium of the internal state to maintain life through biochemical reactions. These physiological responses are independent of other processes such as pain, sentience, or suffering (although they may be associated with them), but any deviation from baseline conditions is indicative of imbalances that may result in adverse conditions for animals. Complexity increases in this sense, since the physiological stress responses are not only taxa-specific, but may also be species-specific. Therefore, it is complicated to search for universal patterns that can serve to establish common bases for all aquatic animals, although it is not impossible. Previously, some authors have addressed and reviewed the welfare of aquatic animals from different points of view [1,2,3,6,20,21,22,23], although we consider that the animal physiology has not been given the importance it deserves. These studies were mostly focused on: (i) Establishing analogies between human and animal brains (looking for shared mental capabilities); (ii) behavioral changes; (iii) evidence of sentience-pain-suffering; and (iv) cortisol or lactate production as the most relevant physiological parameters in teleosts; or (v) mobilization of energy metabolites in other taxa. We observed a lack of consistency in the comparative assessment of welfare between aquatic animals. Physiology may be considered as the common ground to evaluate welfare through the assessment of stress responses. In this regard, in the following sections we define basic concepts of stress physiology and, subsequently, highlight some specific details among different taxa.

### 1.2. Stress Physiology

Animals have evolved maintaining internal body fluids within a controlled range of certain parameters, including pH, ion concentration, oxygen supply, or available energy metabolites, amongst others (see Table 1). All of them are regulated through biochemical reactions involving enzymes, hormones, transporters, and specific proteins or lipids, influenced by temperature. 

The maintenance of this homeostasis after compromising stressful events requires a synchronized action of allostatic changes [48], that enable the return to optimal physiological levels for the animal [49]. These modifications, or physiological stress responses, are grouped into primary, secondary, and tertiary responses (Figure 1) [50]. After integrating a stressor, the animal processes the information through its nervous system and alerts the body. The first responses include the secretion to the circulatory system of neuroendocrine hormones such as catecholamines [51,52] and others, including corticosteroids in vertebrates [21,53,54] and hyperglycemic hormone in crustaceans [55]. These hormones induce secondary stress responses, increasing heart and respiration frequency rates, and mobilizing energy metabolites to cover the demand for energy and oxygen imposed by the stressor [21,31,56,57,58]. If the stressor persists over time, a series of tertiary stress processes will develop that can lead to the collapse of energy stores and the immune system, behavior and reproduction impairments, and eventual death of the animal [21,58,59]. All these responses depend on the taxonomic group, and large differences exist between terrestrial and aquatic animals, especially invertebrates and fish. As an example of these differences, most of these aquatic animals are not able to maintain their internal temperature constant by endogenous means, which can influence other physiological mechanisms related to homeostasis. Therefore, we describe the major physiological differences among crustaceans, cephalopods, elasmobranchs, teleosts, and dipnoans.

### 1.3. Areas of Interest

All information on aquatic animal welfare is useful for aquaculture, fisheries, research, and exhibition. Regardless of the specific objectives of each of these areas, improving animal welfare has a series of clear advantages. Aquatic animals living in adequate conditions present better growth rates and feed optimization and are less prone to diseases, amongst other benefits [61]. These factors are of paramount relevance in aquaculture, where the benefits come from having lower expenses, including feeding and treatment of diseases [62,63]. In fisheries and aquaculture, attention must be paid to the slaughter of animals, since it has been seen that the quality of the meat decreases when the animal is stressed before its death [64,65,66]. Moreover, in catch-and-release fisheries, it is important to minimize stress on animals so that they can fully recover from the process and return to their natural environment [31,57,60,67,68]. Keeping aquatic animals in the best possible conditions during research experiments is mandatory, since the results may be affected by the care conditions [69,70,71,72]. Finally, the legislation seems to be less strict in aquatic organisms that serve as pets or are used in exhibition centers, such as aquaria. However, keepers of ornamental animals benefit from animal welfare knowledge, since their purpose is to maintain organisms in captivity so that they show their best performance, as well as natural behaviors [73]. All of these aspects are greatly affected by stress and the physiological responses described above. At this point, it is important to know the main physiological characteristics of taxa, as well as how stress responses differ among taxonomic groups, to properly recognize stressful situations and be able to act accordingly.

## 2. Taxonomic Differences

### 2.1. Crustaceans

Although there is an arduous debate about whether crustaceans feel pain and suffering, and whether their welfare should be regulated through legislation resulting from consumer demand [74], there are a number of tools to assess their stress physiology [57]. Due to the vast industry associated to crustaceans (mostly aquaculture and fisheries), some issues are worth considering regarding the maintenance of physiological homeostasis, if only to improve the economic performance of the companies. Crustaceans show a complex endocrine regulation that includes the crustacean hyperglycemic hormone (CHH), which regulates various aspects of growth, reproduction, and metabolism [75,76]. Short- and long-term changes in environmental conditions may result in endocrine disruptions that modify the cascade of physiological responses these hormones regulate. Thus, it has been described that inadequate maintenance conditions can lead to impaired growth of animals [77]. Acute stress responses also modify circulating CHH levels as a primary stress response [78]. Secondary stress responses include mobilization of energy metabolites from storage tissues to the circulatory system [57,79], increasing lactate due to anaerobic metabolism of carbohydrates and thus modifying hemolymph pH [55,79], and O_2_/CO_2_ transport through changes in hemocyanin concentration and/or affinity to these gasses [57,80,81]. Perceived stressors also induce changes in the innate immune system of crustaceans [42,57], leading to an overload of the system if the situation is prolonged in time, which may cause death of the animal. Moreover, some decapods seem to perceive threat and react by triggering the described physiological stress responses even if the threat is not real [82]. This information could serve for future debates about suffering in these animals, although our purpose here is simply to provide physiological evidence supporting homeostasis imbalances that may lead to individual health and growth issues, all related to welfare.

### 2.2. Cephalopods

The central nervous system of cephalopods is more developed than that of other invertebrates, and it is currently considered that they experience pain, suffering and anguish [20]. For these reasons, in certain countries such Australia, Canada, New Zealand, Norway, Switzerland, and in the European Union, this group of mollusks has been included in the legislation that regulates welfare in animal experimentation [6]. This situation can lead to confusion and, unintentionally, erroneously (or at least not scientifically proven) imply physiological similarities with vertebrates. For example, a number of studies described physiological responses to steroid hormones (sexual hormones and corticosteroids) in mollusks. However, this is under debate because steroids are not produced by invertebrates [83,84]. What has been shown is that cephalopods react to stressful situations by secreting neuroendocrine messengers (including catecholamines, such as noradrenaline and dopamine) in the hemolymph after an acute-stress [51]. Stress hormones in mollusks, as seen in vertebrates, can have both enhancing and suppressing effects on immune function [85]. Therefore, exposure to heavy metals or to capture induces the activation of the phenoloxidase activity [86], but may have a stimulatory/inhibitory/neutral influence in other innate immune responses including lysozyme, protease, antiprotease, or peroxidase activities depending not only on the stressor but also on the cephalopod species [31,87,88]. Environmental factors, such as temperature, highlight the importance of oxygen supply in these taxa, and the pigment hemocyanin as a critical element in this process [89,90]. Moreover, hemocyanin concentration is also modified in stressful situations and serves as a stress biomarker in cephalopods [31,91]. Biochemical composition of the animals varies with their life stage, and some changes are observed in their storage tissues throughout the year, due to reproductive performance [92]. However, constant hemolymph pH and glucose levels (among other undescribed parameters) seems to be essential for the maintenance of metabolic homeostasis [31], as described in vertebrates. Cephalopods rely on amino acids as the main energy source, although stored glycogen is also relevant, and are mobilized after a stress challenge [31,33,93]. It should be mentioned, as an important difference compared to vertebrates, that acute-stress challenges do not increase lactate (neither in hemolymph nor in muscle) in cephalopods as part of the secondary stress responses [31]. The latter information paves the way for the search for alternative physiological parameters for the evaluation of stress in these animals. Finally, a very interesting methodology to evaluate the physiological state of cephalopods could include the analysis of their dermal mucus as a non-invasive technique, similar to those used with teleost fish [68,94].

### 2.3. Elasmobranchs

The interest in observing the welfare of elasmobranchs (sharks and rays) lies in three specific areas: Research, fisheries, and exhibition of captive animals. These are primitive vertebrates that are of great scientific interest for the study of evolution of certain physiological strategies, such as the specialization of steroid hormones [95]. Moreover, as key elements of marine (and some freshwater) ecosystems, it is important to maintain their biodiversity at healthy levels [96,97]. Some countries have signed binding agreements to release sharks captured as bycatch in certain fisheries [98,99], and efforts are being made to minimize the damage suffered by the fishing process. The physiological recovery of sharks is involved in the improvement of the survival rates after catch-and-release [60]. Recent advances in elasmobranch physiology highlight the release of catecholamines (adrenaline and noradrenaline) and corticosteroids (1α-hydroxycorticosterone) into the bloodstream as primary stress responses [53,100]. Secondary stress responses include changes in circulating ion concentrations [101], pH, and urea as the main metabolite controlling plasma osmolality levels in marine elasmobranchs [60,102]. While amino acids are conspicuously of paramount importance as oxidative substrates in white muscle of this taxa [103], carbohydrates have been shown to be relevant energy substrates after acute stress challenges [53]. The analysis of certain blood parameters seems to offer a picture of the physiological status of elasmobranchs at a certain moment, highlighting plasma pH, glucose, lactate, and K^+^ levels as selected biomarkers. However, large differences are described depending on the species [53,60,68,99,104], so that further studies are needed to better describe short- and long-term stress responses of these animals and their relationship to elasmobranch welfare.

### 2.4. Teleosts

The stress physiology of teleost fish is well characterized compared to other aquatic taxa. Here, we summarize some physiological responses that teleosts experience under stressful conditions, and their utility as a tool to assess welfare. As introduced in Section 1.2, the endocrine cascades that include hypothalamus, pituitary, and chromaffin or interrenal tissues, control the stress physiology in teleosts [105]. Once the CNS perceives the stressor, catecholamines and cortisol are released into the bloodstream as primary stress responses, although their concentrations vary depending on the species [21]. In short-term conditions, the secondary responses elicited by these hormones are mainly directed towards energy allocation to deal with the stressor and, among others, induce plasmatic hyperglycemia and the increase in other metabolites, such as lactate and proteins. Taken together these responses result in increased heart rate, gill vascularization, and hydro-mineral imbalances [21]. If the stressor persists over time, tertiary responses affect the organism at a complex level [21]. As a consequence, teleosts experience metabolic disorders, lower growth rates, immune-deficiencies, impaired development, reproductive disruptions, or alteration of behavioral and social skills, that clearly compromise their welfare [21].

Special importance has been given to teleost aquaculture since there are multiple factors that can cause stress to animals. Impaired welfare conditions reduce growth and induce pathogen diseases, with the consequent reduction in benefits [106]. The literature on physiological stress biomarkers in different cultured species and husbandry conditions is extensive and constantly increasing. Classical stress parameters are related to the effects derived from cortisol actions [50,61,107], and heat shock proteins as cellular responses [108]. Hypoxia and air exposure due to handling are common acute stress situations in aquaculture. For example, gilthead seabream (*Sparus aurata*) exposed to air for three minutes increased plasma catecholamines (adrenaline and noradrenaline) and cortisol, and altered the expression of genes involved in the endocrine response to stress [109]. A similar response was also shown in meagre (*Argyrosomus regius*) in hypoxia and netting stress situations, where higher plasma levels of cortisol, glucose, lactate, and proteins were described [110]. Current approaches tend to assess physiological stress through non-invasive methodologies. In this regard, a strong correlation exists between some biomarkers observed in blood plasma and dermal mucus [110]. Similarly, cortisol can also be measured in gills, scales, and feces [111]. Immune-system parameters have also been shown to be good stress biomarkers in dermal mucus [112]. IgM levels, peroxidase, protease, and antiprotease activities in skin mucus from *S. aurata* reflected the stress responses of crowding and hypoxia conditions [113]. Stunning and slaughtering processes are also observed to affect fish physiology. As an example, CO_2_ narcosis and ice-slurry reduce glucose, lactate, or cortisol levels, in comparison to asphyxia in the European seabass (*Dicentrarchus labrax*) [114]. The stunning method and the conditions prior to slaughtering affect anaerobic metabolism of muscle and induce changes in its pH, compromising the flesh texture and quality, which may have detrimental effects on the economy of this industry and the welfare of these animals prior to euthanasia [66,115].

The assessment of intermediary metabolism is also an important tool to assess welfare in teleost aquaculture. Carbohydrate, lipid, and amino acid management is directly related to both acute and long-term stress situations [116,117]. Therefore, transport processes boost glycogen consumption, glycolysis, and gluconeogenesis pathways in the liver of *S. aurata* [116]. This process also modulated lipid metabolism in the liver during and after recovery from the stress situation, as it was reflected by triglycerides levels and GPDH and HADH enzyme activities [116]. Similarly, the European eel (*Anguilla anguilla*) [118] and the African catfish (*Clarias gariepinus*) [119] showed higher concentrations of plasma non-esterified fatty acids after transport. An inadequate stocking density is probably the most common long-term stress situation in teleost aquaculture. Both low [120] and high stocking densities [117] can induce tertiary stress responses that reduce growth. This effect is directly related to the allocation of energy and the consequent consumption of stored metabolites in muscle. This reduction of welfare conditions is reflected by the inhibition of the GH/IGF system [121], and by an enhanced catabolism of lipids and amino acids [117]. Stocking density can also modulate other cellular and immune responses [122,123]. Therefore, oxidative stress responses in plasma (i.e., CAT, SOD, GR, GST), and Hsp70 expression in brain, liver, and kidney can be used as biomarkers of long-term stress situations [122]. Alternatively, key components of the innate immune system (i.e., IL-1β, LZM, TNF, TLR-3, and MHC) determined in fish skin mucus also reflected the negative effects of long-term impaired welfare [123]. However, stocking density-derived stress in aquaculture must be assessed cautiously, as the responses are different among species and depend on the rearing system [124]. 

In the case of fisheries, the literature on physiological responses remains limited in comparison to aquaculture. The effect of capture, in terms of gear type, capture duration, or emersion time, has been described for some species and cases [61,125]. For example, angling induces lactate and glucose increases in plasma of Japanese meagre (*Argyrosomus japonicus*) [126], while it also increases cortisol and plasma osmolality in Southern bluefin tuna (*Thunnus maccoyii*) [127]. Similarly, angling time in white marlin (*Kajikia albida*) was associated with hematocrit, and Na^+^, K^+^, Mg^2+^, Ca^2+^, and Cl^−^ changes in the blood [128]. A recent study simulated crowding stress during seine capture of Atlantic mackerel (*Scomber scombrus*), showing physiological changes related to anaerobic metabolism in muscle, affecting flesh quality [129]. A similar effect was also described in Atlantic cod (*Gadus morhua*) after trawling [130]. These studies show the relevance of assessing welfare in fisheries from a physiological point of view, as capture conditions affect flesh quality and may have an impact on their final sale. 

The need for physiological tools to better assess aquatic animal welfare should also be taken into account in other fields, including ornamental fish exhibition in public aquaria. Caretakers make great efforts to create the best artificial environments and guarantee fish welfare by controlling water quality parameters. However, external factors such as the unavoidable noise of visitors can increase cortisol levels, as has been described for the lined seahorse (*Hippocampus erectus*) [131]. It is noteworthy that apart from the study of classical biomarkers [61,132], recent reviews addressed new approaches using transcriptomics, proteomics, and metabolomics to assess long-term welfare of teleosts [133,134].

### 2.5. Dipnoans

The Dipnoi belong to the group of sarcopterygian fish, also known as lungfish. These species are considered the closest living relatives of the tetrapods [135]. In this regard, the study of their physiological strategies is important to understand the evolution of terrestrial vertebrates. As a link between early vertebrates (including teleost fish, such as the widespread scientific model zebrafish (*Danio rerio*) and amphibians, lungfish show similarities between both groups. For example, the South American lungfish (*Lepidosiren paradoxa*) contains both cortisol and aldosterone in its blood [136]. As described before, cortisol is considered a primary stress response hormone in teleosts (acting both as a gluco- and a mineral-corticoid), while aldosterone is typical of amphibians and reptiles (and absent in teleosts, elasmobranchs, and agnathans) [95]. All six species of living lungfish are obligate air breathers, and have the ability to enter a prolonged estivation period during the dry season [137,138]. Aldosterone, which is a mineralocorticoid hormone in terrestrial vertebrates related to Na^+^ homeostasis, seems to decrease not only sodium but also glucose circulating concentrations in the estivating African lungfish *Protopterus annectens* [139]. These responses mediated by aldosterone levels have also been described in starving frogs [139]. However, there is no evidence to date describing glucocorticoid responses in these species. It could be assumed that cortisol is the main glucocorticoid hormone in lungfish due to its high plasma concentrations, although they also produce corticosterone, cortisone, 11-deoxycortisol, 11-deoxycorticosterone, and 11-dehydrocorticosterone [136]. However, there remains a lack of knowledge about which hormone is the main manager of energy metabolism in these species. Moreover, although it has been described that African lungfish rely on both carbohydrate and amino acid stores during estivation [140,141], there are no studies describing secondary or tertiary stress responses in this evolutionarily interesting taxonomic group. Unravelling physiological responses to stress in lungfish could serve to better understand the basis of aquatic animal welfare by acting as a link with terrestrial vertebrates, of which there is plenty of literature.

## 3. Future Approaches

It is clear that some sectors of society demand a better quality of life for all aquatic animals that interact with humans. In the case of fisheries and aquaculture, consumers are demanding more (environmentally) sustainable and fair-trade products, which also includes animal care. There are interesting alternatives to improve/mitigate physiological stress responses in aquaculture, such as the use of natural antioxidants [142], and other dietary additives [117,143]. Similarly, the use of essential oils from plants with sedative properties seems to attenuate the physiological responses inherent to fish handling and cultivation [144]. However, the use of these natural compounds, as well as classical synthetic anesthetics, can evoke additional physiological side-effects, including alterations in energy management [145] and oxidative stress status [146]. This shows that some techniques used to reduce stressful situations in aquatic animals may exert physiological responses that impair their welfare status. In any case, best caring/rearing conditions, environmental enrichment and humane slaughter should be validated through in-depth study of their physiological consequences.

Future regulations of aquatic animals´ welfare will need holistic approaches, including proper descriptions of their physiological status. Studies with only few classical stress responses should be viewed with skepticism, especially those in which glucocorticoid levels are assessed as single biomarkers of welfare. These hormones show diurnal and seasonal fluctuations that can lead researchers to unavoidable misinterpretations if they do not consider the regular homeostasis of the animal. The interpretation of glucocorticoid levels gain complexity with the variability derived from interspecific differences between individuals, including “bold” and “shy” animals. Currently, there are some knowledge gaps surrounding the physiological responses to stress in crustaceans, cephalopods, and fish. Furthermore, depending on the taxa, the basal homeostasis remains unknown. For this reason, and as an analytical approach, the processes that trigger homeostatic imbalances should be studied. It is necessary to pay special attention to the sampling methodologies, as the process itself can be stressful. With the exception of teleost fish, the primary responses to stress are practically unknown in the rest of the taxa reviewed here (crustaceans, cephalopods, elasmobranchs, and dipnoans), and what is known about the secondary responses is limited (Table 2). Future steps should include studying these responses to stressful situations in order to unravel common patterns within each taxonomic group. In this way, humans could anticipate situations that compromise the wellbeing of aquatic organisms. The development of non-invasive, simple, and fast physiological techniques is necessary for a future scenario of full coexistence between humans and animals, safeguarding the welfare of all.

## 4. Conclusions

To assess the animal welfare of aquatic species, physiology is an important tool to consider. Physiological responses to suboptimal conditions are widespread among taxa, and involve changes in the energy management and the immune system. Although these responses vary between species, this review gathered the most relevant parameters related to stress in crustaceans, cephalopods, elasmobranchs, teleosts, and dipnoans. This knowledge can be useful for the management of human activities that involve the use of live aquatic animals. 

## Figures and Tables

**Figure 1 biology-10-00061-f001:**
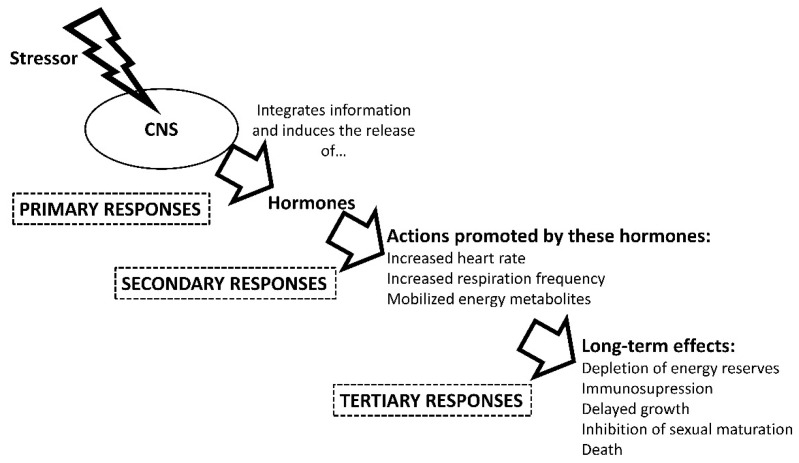
Physiological stress responses in aquatic animals. CNS (Central Nervous System). This figure is based on information from several sources, including [31,50,53,57,58,59,60].

**Table 1 biology-10-00061-t001:** Main physiological parameters of aquatic animals’ homeostasis. It should be noted that certain differences exist depending on the taxa.

System	Parameters	References ^1^
Acid-base balance	H^+^, OH^−^, HCO_3_^−^, PO_4_^2−^, SO_4_^2−^	[24,25]
Hydric-ionic balance	H_2_O, osmolality, Na^+^, Cl^−^, K^+^, Ca^2+^, Mg^2+^, others	[26,27,28,29]
O_2_ (CO_2_) transport	Hemoglobin/hemocyanin, hematocrit	[30,31,32]
Energy management	Glucose, lactate, amino acids, triglycerides, free fatty acids, etc.	[33,34,35]
Immune system (Innate)	Physical barriers, cell-mediated defense (phagocytosis), humoral defense (antimicrobial enzymes, non-specific proteins, complement system), inflammation	[36,37,38,39,40,41,42]
Immune system (Adaptive) ^2^	Cell-mediated defense (B- and T-lymphocytes)	[38]
Free radicals balance	Oxidative stress system	[43,44,45,46]
Others	Hormones, temperature, etc.	[43,47]

^1^ These are a selected group of scientific publications, mostly reviews. ^2^ Adaptive immune system is only present in vertebrates.

**Table 2 biology-10-00061-t002:** Summary of selected useful physiological parameters to assess stress responses and welfare of aquatic animals among different taxa (crustaceans, cephalopods, elasmobranchs, teleosts, and dipnoans).

Taxonomic Group	Parameters	References ^1^
Crustaceans	Crustacean hyperglycemic hormone (CHH).	[75,76,78]
	Hemolymph pH, hemocyanin, glucose, lactate.	[55,57,79,80,81]
	Innate immune parameters (granulocytes, proPO, peroxidase or lysozyme activities).	[42,57]
Cephalopods	Neuroendocrine factors (noradrenaline, dopamine).	[51]
	Innate immune parameters (PO-like, proteases, antiproteases, peroxidase or lysozyme activities).	[31,86,87,88]
	Hemolymph pH, hemocyanin.	[31,89,90,91]
	Glucose, glycogen, amino acids, NOT lactate.	[31,33,92,93]
	Dermal mucus parameters (glucose, lactate, pH).	[69,94]
Elasmobranchs	Catecholamines (adrenaline and noradrenaline).	[100]
	Corticosteroids (1α-hydroxycorticosterone).	[53]
	Plasma pH, osmolality, ions, energy metabolites.	[53,60,101,102]
	Muscle amino acids and carbohydrates (glycogen).	[103]
Teleosts	Neuroendocrine factors (CRH, TRH, POMCs, etc.).	[114,121]
	Plasma catecholamines and cortisol.	[21,50,61,107,109]
	Cortisol in gills, scales or feces.	[111]
	Acid-Base balance.	[66,115]
	Hydro-mineral imbalances.	[128]
	Plasma hematocrit and energy metabolites.	[110,126,127]
	Plasma oxidative stress (CAT, SOD, GR, GST, etc.).	[122]
	Cellular parameters (Hsp70) in brain, liver and kidney.	[108,122]
	Mucus cortisol and energy metabolites.	[110]
	Plasma and skin mucus innate immune parameters.	[112,113,122,123]
	Intermediary metabolism in liver and muscle.	[116,117,118,119]
	Growth rate, condition index, hepatosomatic index.	[106,117,120]
Dipnoans	Glucocorticoids (cortisol, corticosterone, cortisone, 11-deoxycortisol, 11-deoxycorticosterone and 11-dehydrocorticosterone) ^2^.	[136]
	Mineralocorticoids (aldosterone) ^2^.	[136,139]
	Ions, carbohydrates and amino acids.	[139,140,141]

^1^ These are a selected group of scientific publications, mostly reviews. ^2^ Note that the role of this hormones is still not clear in this taxonomic group.

## Data Availability

No new data were created or analyzed in this study.

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
