# Peer review of "Physiology: An Important Tool to Assess the Welfare of Aquatic Animals"

_biology, 2021, doi:10.3390/biology10010061_

Round 1
Reviewer 1 Report
In this review, the authors took up the challenging task, summarizing the available information about the assessment of welfare in aquatic animals. Due to the constantly growing importance of aquaculture production, this is a very critical, yet often neglected topic, deserving the attention of the scientific community.
The text is very well structured, citing a number of recent publications dealing with the assessment of welfare in aquatic animals. Particularly noteworthy is the comparative perspective of the review, evaluating the commonalities between the invertebrates and vertebrates. The review is complemented by informative tables and figures.
In their effort, authors not only summarize the available knowledge but also aimed to provide some future perspective on the use of physiological parameters to assess stress responses and welfare in individual taxa, providing a basic framework for future investigation.
Naturally, the recent progress in the understanding of stress physiology in various species provides an overwhelming amount of information, which could be summarized in a separate review for each studied group of organisms. Yet, I applaud the authors' effort to provide a comprehensive comparative view of this issue and would support the publication of the manuscript in Biology.
Author Response
To facilitate reviewers’ considerations, the modifications in the manuscript are marked in green.
Review Report 1
In this review, the authors took up the challenging task, summarizing the available information about the assessment of welfare in aquatic animals. Due to the constantly growing importance of aquaculture production, this is a very critical, yet often neglected topic, deserving the attention of the scientific community.
The text is very well structured, citing a number of recent publications dealing with the assessment of welfare in aquatic animals. Particularly noteworthy is the comparative perspective of the review, evaluating the commonalities between the invertebrates and vertebrates. The review is complemented by informative tables and figures.
In their effort, authors not only summarize the available knowledge but also aimed to provide some future perspective on the use of physiological parameters to assess stress responses and welfare in individual taxa, providing a basic framework for future investigation.
Naturally, the recent progress in the understanding of stress physiology in various species provides an overwhelming amount of information, which could be summarized in a separate review for each studied group of organisms. Yet, I applaud the authors' effort to provide a comprehensive comparative view of this issue and would support the publication of the manuscript in Biology.
We would like thank the Reviewer to find our review interesting, and appreciate the effort done to summarize all the information available about the physiology of aquatic animals. The objective was to provide a practical document for researchers not familiar with stress responses to consider these analytical tools in the assessment of welfare in aquatic animals. We agree, the comparative view was quite difficult to address, as vertebrate taxa are far more studied than invertebrates. However, as the reviewer commented, we think the interesting similarities described could serve as a basic framework for further studies. Thanks again for your considerations.
Reviewer 2 Report
This manuscript presents a descriptive literature review on the use of physiology to assess the welfare of wild and captive aquatic animals (fishes, sharks, crustaceans, cephalopods, lungfish). Please see my comments below for recommendations on improving the manuscript.
Overall Comments:
- This descriptive literature review lacks a specific objective and as such the author’s purpose for doing the review and how it will advance the science and practice of animal welfare are not clear.
- Early in the manuscript the authors should summarize previous reviews on assessing welfare of aquatic animals and identify how their review differs from previous review.
Specific Comments
- line 11: The accuracy of this statement is highly questionable. I concur that animal quality of life is a concern for some humans, but not all and not necessarily the majority. Either delete or revise.
- line 13: what issue? Be specific
- line 30-31: rewording suggestion: …show the potential of the physiology of crustaceans, cephalopods, sharks, fishes, and lungfish to serve as indicators of their…
- line 46: delete “to describe”
- line 50: this sentence needs some supporting citations for this claim
- line 52: replace “land-based activities” with “terrestrial ecosysystems” and replace “aquatic environments” with “aquatic ecosystems”
- line 56: what does “subjected to human influences” mean? Animals in captivity? Animals managed by humans (i.e., sport fish stocks) or both or something else. Revise to clarify
- line 78: animal exhibition? Is this zoos and nature centers? Or traveling events like circuses that have performing animals? Or all three?
- line 80: in the last years? Be specific and state how many of the last years it has been under revision.
- line 80: delete “other”
- line 84-85: in line with what? Be specific
- line 87: replace “an animal taxon” with “taxa”.
- line 96: insert “terrestrial” between “observe” and “animals”
- line 96-97: replace “species with which we work” with “terrestrial species in captivity or being managed by natural resource agencies.”
- line 97: need clarify that this sentence focuses on aquatic species.
- line 97: I question this statement – there is a large research effort in aquatic ecology that involves aquatic animal behavior in the lab and the field and many creative methods to observe aquatic animal behavior.
- line 103: replace “These responses” with “These assessments” and delete “not entirely clear processes such as”
- line 108: replace “bases” with “basis”
- line 110 and throughout manuscript: replace “taxon” with “taxa”
- line 120-121: allostatic changes in response to what? Seems like there are some important details missing here
- subsection 1.2: somewhere in here I think it is important that the authors discuss how the physiology of these aquatic animals differs with respect to terrestrial animals. Particularly, it should be noted that fishes, sharks, and lung fishes are cold blooded animals and do not internally regulate their body temperatures. This is big physiological difference and potentially could influence other physiological mechanisms related to homeostasis.
- line 132: revise as follows: …differences among crustaceans,….
- line 138: delete “various human activities, including”
- line 141: need supporting citation
- line 145: delete “moments”
- line 148: revise as follows: …since the research results may be affected…
- line 150: replace “exposed” with “displayed”
- line 152: replace “color liveries” with “coloration” or “color patterns”
- line 183: which countries? Be specific.
- line 185-186: this sentence is confusing and needs to be clarified what incorrect similarities are established?
- line 186-187: this sentence contradicts itself and as such needs to be revised.
- line 250: replace “wide” with “large”
- line 286-288: Need to explain further for the non-ichthyologists. I as someone who works with fish understand that schooling fish will not likely exhibit stress responses to being held in high densities in aquaria, but solitary fish would be.
- line 289: This statement is not accurate because there is a lot of work on ecophysiology in fishes has been conducted with sportfishes and game fishes that are the recreational fisheries in many areas of the U.S.
- 35.line 332-333: The accuracy of this statement is highly questionable. I concur that some humans demand a higher quality of life for animals, but not all and not necessarily the majority. Either delete or revise.
- 36.line 348: this statement would be strengthen by adding examples of specific types of physiological biomarkers that should be assessed.
- Table 2: This table is hard to read and needs blank rows between each taxa and their associated physiological biomarkers because I could not tell which biomarkers were associated with each taxa
- Table 2 contains quite a few recommended physiological biomarkers. Is it possible to measure all of these? Also, it would help to identify biomarkers that are common or analogous across taxa like hemolymph pH for crustaceans, cephalopods, plasma pH for sharks. Understanding these types of common responses helps identify cross-taxa responses..
- there are a number of citations in the references that cite regulations, laws, etc. I understand these have different format type than the standard journal article. However, as used a reader does not know the source for these regulations and as such if these regulations are on the internet then reporting the website that they are available would be helpful.
Author Response
To facilitate reviewers’ considerations, the modifications in the manuscript are marked in green.
Review Report 2
This manuscript presents a descriptive literature review on the use of physiology to assess the welfare of wild and captive aquatic animals (fishes, sharks, crustaceans, cephalopods, lungfish). Please see my comments below for recommendations on improving the manuscript.
Overall Comments:
- This descriptive literature review lacks a specific objective and as such the author’s purpose for doing the review and how it will advance the science and practice of animal welfare are not clear.
The aim of this review is to highlight the importance of physiology to assess welfare of aquatic animals, as we consider that it is not entirely clear for all people interested in this topic. We want to provide a comparative perspective among different invertebrate and vertebrate taxa, as a basic framework for future studies related to well-being of crustaceans, cephalopods and fish (elasmobranchs, teleosts and dipnoans). Several changes have been done in the manuscript to highlight the aim of this review.
- Early in the manuscript the authors should summarize previous reviews on assessing welfare of aquatic animals and identify how their review differs from previous review.
We appreciate this comment and a specific sentence has been included in the first section (1.1 Welfare). Moreover, along the text, we have modified some parts for clarity.
Specific Comments
- line 11: The accuracy of this statement is highly questionable. I concur that animal quality of life is a concern for some humans, but not all and not necessarily the majority. Either delete or revise.
We agree with the Reviewer and the sentence has been modified.
- line 13: what issue? Be specific
This was revised as: “Welfare of terrestrial animals is well developed due to ease of observation, but it is less well characterized for aquatic animals.”
- line 30-31: rewording suggestion: …show the potential of the physiology of crustaceans, cephalopods, sharks, fishes, and lungfish to serve as indicators of their…
The sentence was rephrased to: “This review aims to show the potential of the physiology of crustaceans, cephalopods, elasmobranchs, teleosts and dipnoans to serve as indicators of their well-being”.
- line 46: delete “to describe”
“to describe” was removed from the text.
- line 50: this sentence needs some supporting citations for this claim
We have included a final statement in the sentence, indicating that these citations are described in the next paragraph.
- line 52: replace “land-based activities” with “terrestrial ecosystems” and replace “aquatic environments” with “aquatic ecosystems”
The sentence was changed as: “Governmental concern about animal welfare, described in relevant policies, is gaining importance, not only for terrestrial ecosystems but also for aquatic ecosystems”.
- line 56: what does “subjected to human influences” mean? Animals in captivity? Animals managed by humans (i.e., sport fish stocks) or both or something else. Revise to clarify
This issue was clarified in text as follows: “However, due to the lack of sufficient scientific evidence, problems have arisen when determining animal sentience for the entire collection of aquatic animals that are subjected to human influence (i.e. aquaculture, fisheries, zoos, etc.)”.
- line 78: animal exhibition? Is this zoos and nature centers? Or traveling events like circuses that have performing animals? Or all three?
The text was revised as follows: “To the best of our knowledge, the current regulation for keeping animals in zoos and aquaria in Europe is the Council Directive 1999/22/EC”.
- line 80: in the last years? Be specific and state how many of the last years it has been under revision.
This was specified as: “This regulation has been under revision during the last decade.”
- line 80: delete “other”
“other” was removed from the text.
- line 84-85: in line with what? Be specific
The sentence was rephrased as: “Similarly, other countries like Australia, Canada, New Zealand, Norway and Switzerland, include not only cephalopods but also decapod crustaceans.”
- line 87: replace “an animal taxon” with “taxa”.
This phrase was rewritten as: “Current approaches to determining whether taxa should be included in animal welfare regulations are based on animal sentience, but this is a complex term to apply for some aquatic taxa, especially to invertebrates.”
- line 96: insert “terrestrial” between “observe” and “animals”
- line 96-97: replace “species with which we work” with “terrestrial species in captivity or being managed by natural resource agencies.”
- line 97: need clarify that this sentence focuses on aquatic species.
All these comments were addressed as: “This has been possible due to the ease with which we can observe these animals, as well as the (relatively) small number of terrestrial species managed in captivity and in natural environments. However, only (relatively) few aquatic species from those kept in captivity can be observed directly, since there is an immense variety of taxa, with different shapes, sizes and biological needs, making it very difficult to describe their behaviour adequately in comparison to terrestrial species.”.
- line 97: I question this statement – there is a large research effort in aquatic ecology that involves aquatic animal behaviour in the lab and the field and many creative methods to observe aquatic animal behaviour.
We agree with this comment. There are plenty of studies focused on fish ethology, especially on zebrafish due to its importance in research, but also on wild and captive species. However, we want to remark the differences among taxa in comparison to terrestrial animals. Trout, sole and Bluefin tuna are teleosts, but they have different behaviours, albeit all share certain physiological responses that can be studied to assess their welfare. There is a great biodiversity of aquatic animals (around 30000 teleosts, 800 elasmobranchs, 6 dipnoans, 67000 crustaceans and 700 cephalopods), of which a large number interact with humans, so there would be interest in their well-being. Assessing their welfare through behaviour requires more time to recognize similarities between them and avoid the study of individual species separately. We also think the field of ethology is extremely relevant to assess welfare, especially as a non-invasive methodology. Moreover, we consider that the combination of ethology and physiology is of great interest to assess animal welfare.
- line 103: replace “These responses” with “These assessments” and delete “not entirely clear processes such as”
We kindly think “assessments” is not correct in these context. It is not related to the evaluation of physiological parameters, but to the endocrine interactions in the organism. We have rephrased this idea as: “The concept of homeostasis is well known in animals, which is the equilibrium of the internal state to maintain life through biochemical reactions. These physiological responses are independent of other processes such as pain, sentience, or suffering…”.
- line 108: replace “bases” with “basis”
As “bases” is the plural of “basis”, we would like to maintain the context of the sentence as it was previously written.
- line 110 and throughout manuscript: replace “taxon” with “taxa”
As for the previous comment, “taxa” is the plural of “taxon”. Due to our non-native English speaking, we don’t fully understand why “taxa” (the plural form) should be always employed as the Reviewer suggested. However, for your consideration, we revised the use of “taxa/taxon” along the text.
- line 120-121: allostatic changes in response to what? Seems like there are some important details missing here
This was rephrased as: “The maintenance of this homeostasis after compromising stressful events requires a synchronized action of allostatic changes [48], returning to optimal physiological levels for the animal [49]”.
- subsection 1.2: somewhere in here I think it is important that the authors discuss how the physiology of these aquatic animals differs with respect to terrestrial animals. Particularly, it should be noted that fishes, sharks, and lungfishes are cold blooded animals and do not internally regulate their body temperatures. This is big physiological difference and potentially could influence other physiological mechanisms related to homeostasis.
We have modified the last part of this section to accommodate the Reviewer´s comment. Now it reads as: “All these responses depend on the taxonomic group, and great differences exist between terrestrial and aquatic animals, especially in invertebrates and fish. As a crude example to highlight these differences, it should be noted that most of these aquatic animals are not able to maintain their internal temperature constant by endogenous means, which can influence other physiological mechanisms related to homeostasis. Therefore, we describe the major physiological differences among crustaceans, cephalopods, elasmobranchs, teleosts and dipnoans”.
- line 132: revise as follows: …differences among crustaceans,…
This was revised as suggested: “All these responses depend on the taxonomic group; therefore, we describe the major physiological differences among crustaceans, cephalopods, elasmobranchs, teleosts and dipnoans.”.
- line 138: delete “various human activities, including”
This was removed as suggested.
- line 141: need supporting citation
We have included supporting information.
- line 145: delete “moments”
This was removed as suggested.
- line 148: revise as follows: …since the research results maybe affected…
This was rewritten as follows: “Keeping aquatic animals in the best possible conditions is mandatory, since the research results may be affected by the care conditions”.
- line 150: replace “exposed” with “displayed”
This was rewritten as follows: “Finally, the legislation seems to be less strict in aquatic organisms that serve as pets or are used in exhibition centers, such as aquaria.”
- line 152: replace “color liveries” with “coloration” or “color patterns”
This was rewritten as follows: “However, keepers of ornamental animals benefit from animal welfare knowledge, since their purpose is to maintain organisms in captivity so that they show their best performance, as well as natural behaviors”.
- line 183: which countries? Be specific.
This was specified as: “For these reasons, in certain countries such Australia, Canada, the EU, New Zealand, Norway and Switzerland, this group of mollusks has been included in the legislation that regulates animal welfare next to vertebrates.”
- line 185-186: this sentence is confusing and needs to be clarified what incorrect similarities are established?
We thought it was clear. We were referring to the studies about steroid hormones in cephalopods mentioned afterwards. This statement was rewritten as follows: “This situation can lead to confusion, and establish physiological similarities with vertebrates that are not real (or at least have not been scientifically proven).”
- line 186-187: this sentence contradicts itself and as such needs to be revised.
This sentence was rephrased as: “For example, a number of studies describe steroid hormones in mollusks (sexual hormones and corticosteroids), however it has been demonstrated that steroids are not produced by invertebrates, including cephalopods”.
- line 250: replace “wide” with “large”
This was replaced as suggested.
- line 286-288: Need to explain further for the non-ichthyologists. I as someone who works with fish understand that schooling fish will not likely exhibit stress responses to being held in high densities in aquaria, but solitary fish would be.
This was simplified as follows: “However, stocking density-derived stress in aquaculture must be assessed cautiously, as the responses are different among species and depend on the rearing system.”
- line 289: This statement is not accurate because there is a lot of work on ecophysiology in fishes has been conducted with sport fishes and game fishes that are the recreational fisheries in many areas of the U.S.
We agree, and we specified that this statement regarding fisheries was done “in comparison to aquaculture”.
- line 332-333: The accuracy of this statement is highly questionable. I concur that some humans demand a higher quality of life for animals, but not all and not necessarily the majority. Either delete or revise.
We included “some sectors of society” for clarity.
- line 348: this statement would be strengthen by adding examples of specific types of physiological biomarkers that should be assessed.
These examples are described and summarized in Table 2.
- Table 2: This table is hard to read and needs blank rows between each taxa and their associated physiological biomarkers because I could not tell which biomarkers were associated with each taxa.
Blank rows are included between each taxa. Moreover, converting the original “Word” file to a “PDF” format changed the rows in this table. We apologize for this inconvenience.
- Table 2 contains quite a few recommended physiological biomarkers. Is it possible to measure all of these? Also, it would help to identify biomarkers that are common or analogous across taxa like hemolymph pH for crustaceans, cephalopods, plasma pH for sharks. Understanding these types of common responses helps identify cross-taxa responses.
All shown biomarkers can be measured for each taxa. Common biomarkers such as pH in the circulatory system exist, although we consider not to displayed them separately for clarity.
- There are a number of citations in the references that cite regulations, laws, etc. I understand these have different format type than the standard journal article. However, as used a reader does not know the source for these regulation and as such if these regulations are on the internet then reporting the website that they are available would be helpful.
As the Reviewer suggested, the websites, in which each regulation cited was consulted, were included.
Reviewer 3 Report
The authors showed a comprehensive and interesting review on the physiology of aquatic animals welfare. I agree with the data presented and recommend its publication at its present form.
Author Response
To facilitate reviewers’ considerations, the modifications in the manuscript are marked in green.
Review Report 3
The authors showed a comprehensive and interesting review on the physiology of aquatic animals’ welfare. I agree with the data presented and recommend its publication at its present form.
We would like to thank the Reviewer to find our review interesting, and appreciate the effort done to show in a comprehensive way all the information available about the physiology related to welfare of aquatic animals. Thanks again to consider the review acceptable for publication in Biology journal.
Reviewer 4 Report
This review Is an interesting discussion of the physiological measures used to determine stress in fishes, crustaceans and cephalopods. The authors make the case that there are gaps in our knowledge regarding the stress response of elasmobranchs, crustaceans and cephalopods compared with the valuable empirical evidence already known in teleosts. Stress physiology is incredibly useful for measuring an animal’s responses in a variety of contexts.
However, care must be taken with these measures since, for example, cortisol shows a diurnal fluctuation and often a seasonal fluctuation so sampling should be properly controlled and carefully interpreted.
Further the very act of catching a fish or any other sample is highly likely to be stressful. I think these points should be made in this review. Of course some scientists have validated measurement of cortisol from water samples which is less invasive. Overall this review is well written but some of the ideas need updating.
Many behavioural studies have employed physiological measurements in parallel and with the advent of software which accurately measures behaviour without error and bias – it is therefore objective. Therefore your opinion that behaviour is hindering the advancement of fish or other animal welfare is not correct – you do not evidence this opinion in your text. If anything the aquaculture industry has embraced underwater behavioural observations (e.g. https://doi.org/10.3389/fmars.2020.00645).
I do think physiological measures alongside behavioural analysis are very important but behavioural tools on their own provide an effective means of assessing welfare (e.g. https://doi.org/10.1038/s41598-019-45464-w).
Furthermore the http://fishethobase.net/ catalogues behavioural indicators of a variety of fish species (see https://doi.org/10.3390/fishes4020030). I also don’t agree with the statement that the study of fear, pain and anguish (normally termed distress) are underdeveloped. There have been many advances using fish models in these topics with studies using neuroanatomy, neurobiology, molecular and physiological measures as well as behaviour.
Of course behaviour is unobtrusive and non-invasive but physiological measures are much more difficult to make without stress and as I previously stated careful sampling is needed. Overall I am supportive of publication but I would like to see some revisions that expand or elaborate on the authors’ opinion and some text to update the views expressed. None of these are insurmountable. I have some more specific comments:
Title: I don’t think Physiology is a forgotten tool. Change to “Physiology: an Important Tool to Assess the…”
L14 behaviour is not an obstacle in my view and has provided much progress on understanding the welfare of aquatic animals – please rephrase.
L16 Behaviour is objective- rephrase or delete objective.
Abstract
L27 anguish is usually termed distress. Underdeveloped compared to what – there are numerous studies on fear and pain in fishes as well as cephalopods and now crustaceans?
L29 delete objective.
L35 Dipnoi with a capital?
L41 some scientists have made a career out of defining welfare – perhaps this needs to be rephrased that there are three main definitions and so agreement among scientists can be problematic according to what definition they adopt. Anthropic should anthropomorphic Ref 2 is a whole book – think it would be best to refer to a specific chapter.
L88 replace fuzzy with esoteric. There are more precise definitions of sentience (Broom, 2014 Animal Sentience and Welfare https://www.cabi.org/bookshop/book/9781780644042/ defines sentience elegantly; chapter on Mental Capacities of Fishes by Sneddon & Brown also defines sentience and pain https://www.springer.com/gb/book/9783030310103). Yes governments and legislators do make decisions on what animals to protect on the basis that they are considered sentient and capable of suffering so should be protected – don’t you agree?
L91 I think an army of human pain researchers would not agree with you. The Brown and Sneddon chapter recommended above discusses consciousness in fishes. So what would you propose we do to protect animals and what criteria should be applied? They have a stress response? The references cited here are either whole edited books or a review. I think it would be correct to cite a specific book chapter or an empirical paper.
L99 Many studies have been published on bold versus shy personalities or proactive versus reactive coping styles in fishes. There is very good chapter in reference 2 on this very subject. By our argument of course then physiological measures are not much good either as animals also differ in their stress response?
L104 Pain, suffering, fear and distress are inherently stressful and there are studies published on fishes and crustaceans demonstrating that stress hormones are higher in pain treatment groups. For example Elwood and Adams (Biol Lett 2015 doi: 10.1098/rsbl.2015.0800) or White et al. doi.org/10.1016/j.anbehav.2017.08.017).
L163 I think it should be added that the consumer wants animals held in good welfare and it is good for the animal here.
L183 The European Union and many other countries accept cephalopods experience pain and suffer? Please revise.
L301 Please change caretakers to carers.
Author Response
To facilitate reviewers’ considerations, the modifications in the manuscript are marked in green.
Review Report 4
This review Is an interesting discussion of the physiological measures used to determine stress in fishes, crustaceans and cephalopods. The authors make the case that there are gaps in our knowledge regarding the stress response of elasmobranchs, crustaceans and cephalopods compared with the valuable empirical evidence already known in teleosts. Stress physiology is incredibly useful for measuring an animal’s responses in a variety of contexts.
However, care must be taken with these measures since, for example, cortisol shows a diurnal fluctuation and often a seasonal fluctuation so sampling should be properly controlled and carefully interpreted.
Further the very act of catching a fish or any other sample is highly likely to be stressful. I think these points should be made in this review. Of course some scientists have validated measurement of cortisol from water samples which is less invasive. Overall this review is well written but some of the ideas need updating.
We appreciate the Reviewer´s comments and agree with the idea that there are many different approaches rather than just analysing stress physiology to assess animal welfare. We do not intent to discredit other modalities, but to highlight the usefulness of physiology to evaluate the welfare of aquatic animals.
In order to improve the quality of the text, we included the suggested comments about the fluctuations in some stress-related parameters such as cortisol and the effect of the sampling method, as follows:
“Future regulations of aquatic animals´ welfare will need holistic approaches, including proper descriptions of their physiological status. Studies with only few classical stress responses should be considered with scepticism, especially to those where glucocorticoid levels are assessed as single biomarkers of welfare. These hormones show diurnal and seasonal fluctuations that can lead researchers to unavoidable misunderstandings if they do not look at the regular homeostasis of the animal. This also get more complexity with the variability derived from interspecific differences between individuals, including “bold” and “shy” animals. Currently, there are some knowledge gaps surrounding the physiological responses to stress in crustaceans, cephalopods and fish. Furthermore, depending on the taxa, the basal homeostasis is not even known. For this reason, and as an analytical approach, the processes that trigger homeostatic imbalances should be studied, paying special attention to sampling procedures, as the method itself can be stressful”.
Many behavioural studies have employed physiological measurements in parallel and with the advent of software which accurately measures behaviour without error and bias – it is therefore objective. Therefore, your opinion that behaviour is hindering the advancement of fish or other animal welfare is not correct – you do not evidence this opinion in your text. If anything the aquaculture industry has embraced underwater behavioural observations (e.g. https://doi.org/10.3389/fmars.2020.00645).
We want to apologize if the text gives a wrong idea about the capabilities of analysing animal behaviour as a tool to assess welfare. Our objective is to emphasize that, due to the immense number of aquatic species (around 30000 teleosts, 800 elasmobranchs, 6 dipnoans, 67000 crustaceans and 700 cephalopods), evaluating the behaviour of all of them will take time. Especially since we are still far from knowing common behavioural responses that can be used to assess the welfare of various aquatic species. However, there are many physiological similarities between species (and taxa) that may be of help to evaluate their well-being. Some parts of the manuscript were modified for clarity.
I do think physiological measures alongside behavioural analysis are very important but behavioural tools on their own provide an effective means of assessing welfare (e.g. https://doi.org/10.1038/s41598-019-45464-w).
We agree with the Reviewer and appreciate the indicated manuscripts, since they have an undeniable scientific value.
Furthermore, the http://fishethobase.net/ catalogues behavioural indicators of a variety of fish species (see https://doi.org/10.3390/fishes4020030). I also don’t agree with the statement that the study of fear, pain and anguish (normally termed distress) are underdeveloped. There have been many advances using fish models in these topics with studies using neuroanatomy, neurobiology, molecular and physiological measures as well as behaviour.
Again, the Reviewer is right. The advances in this topic are growing day by day, but we are still far from knowing how crustaceans and cephalopods process and manage fear, pain or anguish.
Of course behaviour is unobtrusive and non-invasive but physiological measures are much more difficult to make without stress and as I previously stated careful sampling is needed. Overall I am supportive of publication but I would like to see some revisions that expand or elaborate on the authors’ opinion and some text to update the views expressed. None of these are insurmountable. I have some more specific comments:
Title: I don’t think Physiology is a forgotten tool. Change to “Physiology: an Important Tool to Assess the…”
As the Reviewer suggested, the title was changed to “Physiology: an Important Tool to Assess the Welfare of Aquatic Animals”.
L14 behaviour is not an obstacle in my view and has provided much progress on understanding the welfare of aquatic animals– please rephrase.
This was rephrased as follows: “Classic methodologies, such as behavioural observation, seem difficult to improve alone the well-being of aquatic animals, mainly due to the large number of species and the complexity to obtain comparative results among taxa”.
L16 Behaviour is objective-rephrase or delete objective.
“Objective” was removed from the text.
Abstract
L27 anguish is usually termed distress. Under developed compared to what – there are numerous studies on fear and pain in fishes as well as cephalopods and now crustaceans?
We agree with this, and “undeveloped” was replaced in the sentence as follows: “While evaluating their emotions (such as fear, pain and anguish) is currently in development for aquatic species, and understanding the natural behaviour of all aquatic taxa that interact with humans is a task that requires more time, the evaluation of internal responses in the organisms can be carried out using analytical tools.”
L29 delete objective.
This was addressed in the previous comment
L35 Dipnoi with a capital?
We apologize for this misspelling. We revised the whole text, and Dipnoi and dipnoans are now properly used and written along the review.
L41 some scientists have made a career out of defining welfare– perhaps this needs to be rephrased that there are three main definitions and so agreement among scientists can be problematic according to what definition they adopt. Anthropic should anthropomorphic Ref 2 is a whole book – think it wouldbe best to refer to a specific chapter.
This was amended in text as follows: “Animal welfare is a difficult concept to define, mostly due to anthropomorphic perception, but it is usually associated with three different points: i) the physiological functioning of an animal; ii) its natural living; and iii) its feelings.”.
L88 replace fuzzy with esoteric. There are more precise definitions of sentience (Broom, 2014 Animal Sentience and Welfare https://www.cabi.org/bookshop/book/9781780644042/defines sentience elegantly; chapter on Mental Capacities of Fishes by Sneddon & Brown also defines sentience and pain https://www.springer.com/gb/book/9783030310103). Yes, governments and legislators do make decisions on what animals to protect on the basis that they are considered sentient and capable of suffering so should be protected – don’t you agree?
We removed the word “fuzzy” for clarity and appreciate the Reviewer´s sense of humor. We wanted to give a subjective point to a scientific text and the Reviewer has done well to point it out to us.
Regarding the use of the word “sentience”: we just expose that physiological parameters can be applied to improve the assessment of welfare among taxa, and can be considered to improve the related legislation. Consciousness of fish, cephalopods and decapods is still a matter of debate, and further studies should be necessary to change the idea shared by many authors that these concept is inherent to humans. Our review is far from wanting to delve into this debate. The text was modified for clarity and now the use of terms such as “consciousness”, “sentience” and “suffering” is more careful to avoid misunderstandings.
L91 I think an army of human pain researchers would not agree with you. The Brown and Sneddon chapter recommended above discusses consciousness in fishes. So what would you propose we do to protect animals and what criteria should be applied? They have a stress response? The references cited here are either whole edited books or a review. I think it would be correct to cite a specific book chapter or an empirical paper.
In the cited paragraph not only the term “pain” was discussed, but also “suffering” and “consciousness”. We unified these concepts in a rude way, since they are used in many cases as arguments to legislate regarding animal welfare.
We want to highlight that stress responses are not only those related to distress, and distress do not refer only to “anguish” (a human concept). Distress responses can be reflected in an impaired growth or condition factor but not necessarily due to anxiety or anguish of the animals. Measuring hormonal levels cannot be assumed as a dogmatic parameter of welfare. We postulate (supported by the cited references) that a set physiological tools and parameters can improve the consideration of animal welfare among the aquatic taxa described, especially in a comparative way. Determining pain in zebrafish may be relatively easy, but its quantification in mackerel captured by seine purse seems more difficult. However, some physiological responses may be shared between taxa, and be of help as a common basis to evaluate stress.
The cited references were revised, as suggested.
L99 Many studies have been published on bold versus shy personalities or proactive versus reactive coping styles in fishes. There is very good chapter in reference 2 on this very subject. By our argument of course then physiological measures are not much good either as animals also differ in their stress response?
The Reviewer is completely right. There is lot work to do in this topic, but this was not the scope of our review. We included this information in the text for clarity.
L104 Pain, suffering, fear and distress are inherently stressful and there are studies published on fishes and crustaceans demonstrating that stress hormones are higher in pain treatment groups. For example Elwood and Adams (Biol Lett 2015 doi:10.1098/rsbl.2015.0800) or White et al.doi.org/10.1016/j.anbehav.2017.08.017).
We do not question these studies, but as we exposed in a previous comment, pain, suffering, and fear, are difficult terms to assess in certain situations.
L163 I think it should be added that the consumer wants animals held in good welfare and it is good for the animal here.
This idea was added in the text as follows: Although there is an arduous debate about whether crustaceans feel pain and suffering, and whether their welfare should be regulated through legislation due to consumers’ demand.
L183 The European Union and many other countries accept cephalopods experience pain and suffer? Please revise.
Yes, they do. You can check the Directive 2010/63/EU. It is also revised in doi:10.1093/icesjms/fsy067.
L301 Please change caretakers to carers.
We prefer to maintain caretakers to be consistent with the American spelling used along the review. This was checked with a native speaker and considered properly employed.
Round 2
Reviewer 2 Report
This manuscript is a revised version of one I reviewed previously that presents a descriptive literature review on the use of physiology to assess the welfare of wild and captive aquatic animals (fishes, sharks, crustaceans, cephalopods, lungfish). Please see my comments below for recommendations on improving the manuscript.
line 15: replace “complexity” with “difficulty”
line 16: replace “define” with “identify”
line 26: insert “methods for” between “While” and “evaluating”
line 46: replace “They” with “These five principles”
line 53: replace “as seen in the next paragraph” with “as discussed below”
line 63: procedures for wild animals, captive animals, or both? Please clarify.
line 75: replace “text” with “legislation”
line 84-85: this sentence is worded odd and could be restructured. Also, it indicates multiple organizations are providing input but only one example is given. I suggest the following rewording: “In the past decade this regulation has been under revision and with input being provided by organizations such as the European Association of Zoos and Aquaria and “insert name of second organization [18].
Line 90: include cephalopods and decapods in what? Need to specify.
Line 97: rewording suggestion: …for animals. Therefore, this….
Line 102: delete “from those”
Line 113-115: This section summarizing how this review differs from previous reviews can be improved to add greater clarity and to highlight the novelty of this review. First, it is unclear what it means that some authors have addressed animal welfare from different points of view. Briefly providing some examples would help clarify this. Also, as written this section suggests that these reviews did not give physiology the attention it deserves. A brief statement or two here specifying the physiology topics discussed in previous reviews papers differs from your review will highlight for the reader the novelty of this review.
Line 128: rewording suggestion: …[48] to enable a return to optimal…
Line 139: delete “in”
Line 159: insert” during research experiments” after “conditions” and before “is”
Line 166: rewording suggestion: …as well as how stress responses differ among taxonomic groups to properly…
Line 189: replace “show” with “provides” and replace “evidences” with “evidence”
Line 194: avoid use of “EU” acronym and simply write out “European Union”
Line 195-196: what does it mean “legislation that regulates animal welfare next to vertebrates”? This is confusing. Does it mean the cephalopod component of the regulation is next to the legislative text of the vertebrate regulation? Or is it simply that cephalopods and vertebrates are included together in the same legislative text?
Line 196: replace “establish” with “unintentionally imply”
Line 198-199: rewording suggestion: …and corticosteroids). However, it has been…
Line 198-200: I acknowledge that the authors have attempted to address my previous review comment, but these sentences are still contradictory. First it is stated that studies have documented steroid hormones in mollusks then it is stated that steroids are not produced by invertebrates, including cephalopods. So to me this is contradictory and confusing because mollusks and cephalopods are all invertebrates so if as the second sentence suggests – if all invertebrates (which includes mollusks and cephalopods) do not produce steroids, then why are steroid hormones documented in mollusks? Are those mollusk steroid research results wrong? Please revise and clarify to resolve the contradiction in this section.
Line 210: in which species? Mollusks or cephalopods? Need to clarify.
Line 215: delete “in these species”
Line 217: insert “in cephalopods “ after “muscle)” and before “as”. Additionally, if this is true for other invertebrates other than cephalopods then consider adding “in cephalopods and other invertebrates”
Line 221: replace “as is already being done in” with “similar to those used with”
Line 230: replace “correct” with “their”
Line 240: replace “may be necessary” with “are needed”
Line 241: replace “situations in” with “responses of”
Line 276: replace “welfare” with “physiology”. This keeps the readers focus on physiological responses.
Line 280: insert “and the welfare of these animals prior to euthanasia” at the end of this sentence
Line 282-283: replace “the aquaculture of teleosts.” with “teleost aquaculture.”
Line 298: insert “of fish” between “mucus” and “also”
Line 315: insert “aquatic animal” between “assess” and “welfare” and replace “on” with “in”
Line 328: replace “shows” with “contains”
Line 334: decrease concentration of what? Na+ , plasma glucose, or both? Need to clarify.
Line 333-335: This is an interesting observation about similarity between lungfish and starving frogs but the similarity is loss because it is at the end of a long sentence. I suggest splitting this into two sentence with the first establishing how aldosterone changes Na+ and plasma glucose in lungfish. Then adding a second sentence that clarifies that this aldosterone change in lungfish is similar to the stress response exhibited by starving frogs.
Line 350: replace “take into account” with “incorporates”
Line 358: replace “impairing” with “that impairs”
Line 364: replace “considered” with “viewed”
Line 366: replace “misunderstandings” with “misinterpretations” and replace “look” with “consider”
Line 367: what gets more complexity? Need to clarify
Line 372: it is not clear that “sampling procedures” and “methods” are the same thing being referenced here. Replace “sampling procedures’ with “physiological research methods” and “the method itself” with “these methods”
Line 375: delete “a revision of the major physiological responses to stress are shown in”
Line 379: insert “physiological” between “fast” and “techniques”
Table 2, third row crustaceans – replace “some” with “selected”. Also, provide some examples in parentheses here at end of the row like that provided for cephalopods for neuroendocrine factors.
Table 2, second row cephalopods - Also, provide some examples of innate immune parameters in parentheses here at end of the row like that provided for cephalopods for neuroendocrine factors.
Table 2, fifth row cephalopods - Also, provide some examples of dermal mucus parameters in parentheses here at end of the row like that provided for cephalopods for neuroendocrine factors.
Table 2, sixth row dipnoans – delete this row because it does not provide any examples of secondary and tertiary responses.
Author Response
To facilitate reviewers’ considerations, the last modifications in the manuscript are marked in yellow. Changes from original version are still marked in green.
Reviewer 2 Report Round 2
This manuscript is a revised version of one I reviewed previously that presents a descriptive literature review on the use of physiology to assess the welfare of wild and captive aquatic animals (fishes, sharks, crustaceans, cephalopods, lungfish). Please see my comments below for recommendations on improving the manuscript.
We would like to thank the Reviewer for her/his comments. We have addressed all of them point by point in the text.
Line 15: replace “complexity” with “difficulty”
This was replaced as suggested.
Line 16: replace “define” with “identify”
This was replaced as suggested.
Line 26: insert “methods for” between “While” and “evaluating”
This was rephrased as: “While methods for evaluating their emotions (such as fear, pain and anguish) is currently being developed for aquatic species,...”.
Line 46: replace “They” with “These five principles”
“These five principles” was inserted as suggested.
Line 53: replace “as seen in the next paragraph” with “as discussed below”
Replaced as suggested.
Line 63: procedures for wild animals, captive animals, or both? Please clarify.
This was revised as: “Legislation tends to regulate all taxa that are kept captive, as well as their procedures of humane slaughtering.”
Line 75: replace “text” with “legislation”
“text” was replaced for “regulation”
Line 84-85: this sentence is worded odd and could be restructured. Also, it indicates multiple organizations are providing input but only one example is given. I suggest the following rewording: “In the past decade this regulation has been under revision and with input being provided by organizations such as the European Association of Zoos and Aquaria and “insert name of second organization [18].
The sentence was rephrased as follows: “However, the European Commission [17], and the European Association of Zoos and Aquaria (EAZA) [18], have published recent documents of good practices to improve the standards for the accommodation and care of animals in zoos and aquaria.”
Line 90: include cephalopods and decapods in what? Need to specify.
“…in their regulations of animal experimentation” was added to specify this issue.
Line 97: rewording suggestion: …for animals. Therefore, this….
This was revised as suggested.
Line 102: delete “from those”
Deleted as suggested.
Line 113-115: This section summarizing how this review differs from previous reviews can be improved to add greater clarity and to highlight the novelty of this review. First, it is unclear what it means that some authors have addressed animal welfare from different points of view. Briefly providing some examples would help clarify this. Also, as written this section suggests that these reviews did not give physiology the attention it deserves. A brief statement or two here specifying the physiology topics discussed in previous reviews papers differs from your review will highlight for the reader the novelty of this review.
This section was modified, as suggested.
Line 128: rewording suggestion: …[48] to enable a return to optimal…
The phrase was revised as: “The maintenance of this homeostasis after compromising stressful events requires a synchronized action of allostatic changes [48], that enable the return to optimal physiological levels for the animal [49].”
Line 139: delete “in”
Deleted as suggested.
Line 159: insert” during research experiments” after “conditions” and before “is”
The phrase was rewritten as: “Keeping aquatic animals in the best possible conditions during research experiments is mandatory, since the results may be affected by the care conditions [68-71].”
Line 166: rewording suggestion: …as well as how stress responses differ among taxonomic groups to properly…
This was revised as: “At this point, it is important to know the main physiological characteristics among taxa, as well as how stress responses differ among taxonomic groups, to properly recognize stressful situations and be able to act accordingly.”
Line 189: replace “show” with “provides” and replace “evidences” with “evidence”
This was rewritten as: “This information could serve for future debates about suffering in these animals, though our purpose here is just to provide physiological evidences supporting homeostasis imbalances that may lead to individual health and growth issues, all related to welfare.”
Line 194: avoid use of “EU” acronym and simply write out “European Union”
Revised as suggested.
Line 195-196: what does it mean “legislation that regulates animal welfare next to vertebrates”? This is confusing. Does it mean the cephalopod component of the regulation is next to the legislative text of the vertebrate regulation? Or is it simply that cephalopods and vertebrates are included together in the same legislative text?
We have clarified this as: “For these reasons, in certain countries such Australia, Canada, New Zealand, Norway, Switzerland, and in the European Union, this group of mollusks has been included in the legislation that regulates welfare in animal experimentation [6]”.
Line 196: replace “establish” with “unintentionally imply”
Replaced as suggested.
Line 198-199: rewording suggestion: …and corticosteroids). However, it has been…
This was amended in the next comment.
Line 198-200: I acknowledge that the authors have attempted to address my previous review comment, but these sentences are still contradictory. First it is stated that studies have documented steroid hormones in mollusks then it is stated that steroids are not produced by invertebrates, including cephalopods. So to me this is contradictory and confusing because mollusks and cephalopods are all invertebrates so if as the second sentence suggests – if all invertebrates (which includes mollusks and cephalopods) do not produce steroids, then why are steroid hormones documented in mollusks? Are those mollusk steroid research results wrong? Please revise and clarify to resolve the contradiction in this section.
It is a contradiction itself. Some authors described physiological responses in mollusks induced by synthetic steroid hormones in “vertebrate-like” experimental designs. However, the mechanisms to synthetize these hormones were not yet described in invertebrates. We tried to clarify this as: “For example, a number of studies described physiological responses to steroid hormones (sexual hormones and corticosteroids) in mollusks. However, this is under debate because steroids are not produced by invertebrates [82,83]. What has been shown is that cephalopods react to stressful situations by secreting neuroendocrine messengers (including catecholamines, such as noradrenaline and dopamine) in the haemolymph after an acute-stress [51].”.
Line 210: in which species? Mollusks or cephalopods? Need to clarify.
This was clarified in the text: “…and serves as a stress biomarker in cephalopods [31,90]”.
Line 215: delete “in these species”
Deleted as suggested.
Line 217: insert “in cephalopods “after “muscle)” and before “as”. Additionally, if this is true for other invertebrates other than cephalopods then consider adding “in cephalopods and other invertebrates”
We added “in cephalopods”, but we have not considered “and other invertebrates” to avoid misunderstandings.
Line 221: replace “as is already being done in” with “similar to those used with”
Revised in the text as suggested.
Line 230: replace “correct” with “their”
This was rephrased as: “The physiological recovery of sharks is involved in the improvement of the survival rates after catch-and-release”.
Line 240: replace “may be necessary” with “are needed”
This was replaced as suggested.
Line 241: replace “situations in” with “responses of”
This was replaced as suggested.
Line 276: replace “welfare” with “physiology”. This keeps the readers focus on physiological responses.
Thanks for this comment. This was replaced as suggested.
Line 280: insert “and the welfare of these animals prior to euthanasia” at the end of this sentence
We have included that comment in the text.
Line 282-283: replace “the aquaculture of teleosts.” with “teleost aquaculture.”
Replaced as suggested.
Line 298: insert “of fish” between “mucus” and “also”
This was included in the text.
Line 315: insert “aquatic animal” between “assess” and “welfare” and replace “on” with “in”
This was rewritten as follows: “The need for physiological tools to better assess aquatic animal welfare should also be taken into account in other fields, including ornamental fish exhibition in public aquaria.”
Line 328: replace “shows” with “contains”
Replaced as suggested.
Line 334: decrease concentration of what? Na+, plasma glucose, or both? Need to clarify.
This was amended in the next comment.
Line 333-335: This is an interesting observation about similarity between lungfish and starving frogs but the similarity is loss because it is at the end of a long sentence. I suggest splitting this into two sentence with the first establishing how aldosterone changes Na+ and plasma glucose in lungfish. Then adding a second sentence that clarifies that this aldosterone change in lungfish is similar to the stress response exhibited by starving frogs.
This issue was amended as follows: “Aldosterone, which is a mineralocorticoid hormone in terrestrial vertebrates related to Na+ homeostasis, seems to decrease not only sodium but also glucose circulating concentrations in the estivating African lungfish Protopterus annectens [139]. These responses mediated by aldosterone levels have been also described in starving frogs [139].”
Line 350: replace “take into account” with “incorporates”
This was replaced in the text.
Line 358: replace “impairing” with “that impairs”
This was replaced as suggested.
Line 364: replace “considered” with “viewed”
Changed as suggested.
Line 366: replace “misunderstandings” with “misinterpretations” and replace “look” with “consider”
Both words were replaced in the text.
Line 367: what gets more complexity? Need to clarify
This issue was clarified as: “The interpretation of glucocorticoid levels gets more complexity with the variability derived from interspecific differences between individuals, including “bold” and “shy” animals.”
Line 372: it is not clear that “sampling procedures” and “methods” are the same thing being referenced here. Replace “sampling procedures’ with “physiological research methods” and “the method itself” with “these methods”
We have revised this as: “It is necessary to pay special attention to the sampling methodologies, as the process itself can be stressful.”
Line 375: delete “a revision of the major physiological responses to stress are shown in”
Deleted as suggested.
Line 379: insert “physiological” between “fast” and “techniques”
“physiological” was included as suggested.
Table 2, third row crustaceans – replace “some” with “selected”. Also, provide some examples in parentheses here at end of the row like that provided for cephalopods for neuroendocrine factors.
Table 2, second row cephalopods - Also, provide some examples of innate immune parameters in parentheses here at end of the row like that provided for cephalopods for neuroendocrine factors.
Table 2, fifth row cephalopods - Also, provide some examples of dermal mucus parameters in parentheses here at end of the row like that provided for cephalopods for neuroendocrine factors.
Table 2, sixth row dipnoans – delete this row because it does not provide any examples of secondary and tertiary responses.
All suggestions were included.
Reviewer 4 Report
The authors have responded to my comments. Therefore, this review can be published.
Author Response
We appreciate the Reviewer´s effort and guidance.